# Prototype Development of a Temperature-Sensitive High-Adhesion Medical Tape to Reduce Medical-Adhesive-Related Skin Injury and Improve Quality of Care

**DOI:** 10.3390/ijms23137164

**Published:** 2022-06-28

**Authors:** Shawn Swanson, Rahaf Bashmail, Christopher R. Fellin, Vivian Luu, Nicholas Shires, Phillip A. Cox, Alshakim Nelson, Devin MacKenzie, Ann-Marie Taroc, Leonard Y. Nelson, Eric J. Seibel

**Affiliations:** 1Human Photonics Laboratory, Department of Mechanical Engineering, University of Washington, Seattle, WA 98195, USA; sshawn@uw.edu (S.S.); rahafkbashmail@gmail.com (R.B.); vluu2@uw.edu (V.L.); n.shires16@gmail.com (N.S.); lyn3832@uw.edu (L.Y.N.); 2Department of Chemistry, University of Washington, Seattle, WA 98195, USA; fellinchris@gmail.com (C.R.F.); alshakim@uw.edu (A.N.); 3Washington Clean Energy Testbeds, University of Washington, Seattle, WA 98105, USA; coxp51@uw.edu; 4Department of Materials Science and Engineering, University of Washington, Seattle, WA 98195, USA; jdmacken@uw.edu; 5Seattle Children’s Hospital, Seattle, WA 98105, USA; annmarie.taroc@seattlechildrens.org

**Keywords:** medical device, functional polymer, temperature responsive, adhesive film, skin injury

## Abstract

Medical adhesives are used to secure wound care dressings and other critical devices to the skin. Without means of safe removal, these stronger adhesives are difficult to painlessly remove from the skin and may cause medical-adhesive-related skin injuries (MARSI), including skin tears and an increased risk of infection. Lower-adhesion medical tapes may be applied to avoid MARSI, leading to device dislodgement and further medical complications. This paper outlines the development of a high-adhesion medical tape designed for low skin trauma upon release. By warming the skin-attached tape for 10–30 s, a significant loss in adhesion was achieved. A C14/C18 copolymer was developed and combined with a selected pressure-sensitive adhesive (PSA) material. The addition of 1% C14/C18 copolymer yielded the largest temperature-responsive drop in surface adhesion. The adhesive film was characterized using AFM, and distinct nanodomains were identified on the exterior surface of the PSA. Our optimized formulation yielded 67% drop in adhesion when warmed to 45 °C, perhaps due to melting nanodomains weakening the adhesive–substrate boundary layer. Pilot clinical testing resulted in a significant decrease in pain when a heat pack was used for removal, giving an average pain reduction of 66%.

## 1. Introduction

Medical adhesive tapes are an integral part of healthcare delivery and are used in all care settings to cover and secure to the skin wound dressings or critical medical devices such as intravenous (IV) lines [1]. Medical tapes consist of pressure-sensitive adhesive (PSA) applied to a plastic or fabric backing that functions as a carrier for the adhesive, providing structural and protective properties [2]. The combination of different backings and adhesives determines the characteristics of the tape, including levels of adhesion and water resistance, and informs nurses’ decisions in the selection of the appropriate tape for a patient in a specific situation [3]. High-adhesion medical tapes are often used to protect wounds from the environment, and to prevent accidental dislodgement of medical devices. Although effective and easy to apply, the adhesion of these tapes can increase over time, leading to a painful and time-consuming removal process that may ultimately result in tape-induced injuries [3].

Medical-adhesive Related Skin Injury (MARSI) is defined as “erythema and/or other manifestations of cutaneous abnormality that persists 30 min or more after the removal of the adhesive” [2]. These injuries occur when superficial layers of skin are removed by medical adhesives, and include skin tears, blisters, and stripping of the skin [4]. MARSI compromises skin integrity and causes subsequent pain and discomfort. Skin injuries can increase the risk of severe infections, increase wound size, and delay wound healing, all of which significantly impact patient safety and increase medical costs [5].

Ten years ago, there were an estimated 1.5 M MARSI cases annually in the United States [6]. Although MARSI is prevalent, it is underrecognized and underreported [4]. A recent study showed that 98.6% of nurses considered skin tears common or very common [7], with prevalence rates of 13% in the general population [8]. MARSI can occur at any age but is especially ubiquitous in the young and elderly, due to the fragility of their skin. Neonatal skin is around 50% thinner than adult skin, and the epidermal cell layers can strip more easily during tape removal, reducing the skin’s barrier function and resulting in much greater instances of skin trauma [4]. Furthermore, nurses in neonatal intensive care units and nursing homes considered skin tears “extremely common” [2,9]. Higher risks are borne by children fighting disease, which can make a skin tear or infection life-threatening. This is particularly important when the tape is applied and removed repeatedly at the same skin location, such as at an implanted central venous catheter. An observational study examining the occurrence of MARSI in a pediatric intensive care unit for critically ill children reported an incidence rate of over 50% [10]. In addition to causing significant pain, distress, and medical complications, the cost of MARSI includes an average 24 min of extra time, added medical supplies, and a follow-up medical doctor’s visit in over 10% of cases [11]. These effects illustrate both the additional health resources expended, and the patient distress caused by MARSI.

To avoid MARSI, an alternative option is to choose lower-adhesion medical tapes, which increases the risk of critical device dislodgement and is classified as a medical error [2,12]. Among low-adhesion tapes is 3M™’s Kind Removal Tape™ (KRT), which is a silicone-based adhesive that has lower surface tension and maintains a constant level of adherence over time [4]. A comparative study of KRT and standard acrylate-based medical tapes, involving 200 nurses, was conducted over a two-week period [13]. Of the 29% of the nurses that did not prefer the lower adhesion tape, over 75% were dissatisfied with KRT due to unreliable adherence. Furthermore, silicone-based tapes adhere poorly to other silicone products and to plastic tubing [2]. As such, KRT and similar low-adhesion tapes cannot be used to properly secure critical medical devices to the skin, which is essential for patient safety. This deficiency leaves nurses with the unsatisfactory option of high-adhesion acrylate-based medical tapes for many applications, carrying the risk of MARSI during removal.

In other attempts to minimize skin damage and discomfort, medical adhesive removers are often used during the tape removal process [4]. These include alcohol-based, oil-based, and silicone-based removers [2]. However, these removers have limited functionality since they cannot permeate the tape’s backing and must be applied at the adhesive-skin interface while the tape is slowly pulled away from the skin. With cloth-backed tapes, these solvents can soak through the adhesive with time, easing the removal process, but may lead to irritation and leave significant adhesive residue on the skin. Additionally, neonatal skin guidelines advise against using alcohol and organic solvent-based adhesive removers [2]. Therefore, there is a pressing need for a safe and trauma-free adhesive.

A high-adhesion tape that can transition to lower adhesion at the time of removal could greatly reduce MARSI, if not entirely eliminate it, while maintaining a high quality of care. This could be achieved by incorporating a switch mechanism into the adhesive that could be activated by heat or light exposure. Ultraviolet to blue light can create a photoreaction in polymers that can rapidly and dramatically change material properties, as exemplified by the large market for light-activated epoxy resins [14]. This general mechanism of using curable material and photo-initiator combined with PSA has been extended to a new commercial medical tape with switchable adhesion [15]. The release mechanism uses light exposure from a separate device to reduce adhesion.

In our development of ThermoTape™ (previously UnTape), we explored the use of a photothermal switch to release the adhesion of a temperature-sensitive tape developed for non-medical applications [16]. This customized tape system consisted of a near-infrared (NIR) photothermal sensitive tape and a NIR light source with non-contact thermal feedback control. However, requirement for a separate device to remove the adhesive causes logistical problems, would increase costs and regulatory challenges, and was viewed unfavorably by the interviewed nurses.

Thus, we propose the incorporation of temperature-responsive additives (TRAs) as a means of introducing thermal debonding behavior into the adhesive interface between the PSA and the skin or device. These additives can be added directly into the PSA formulation, or they can be selectively applied during manufacture. For skin contact applications, the thermal debonding temperature must not exceed the skin pain threshold temperature (~45 °C) and must be above normal (~35 °C) and febrile (~37 °C) skin temperature [17]. We have developed a mechanism whereby the tape is warmed for less than a minute with a heat pack prior to removal, leading to a rapid and dramatic reduction of the force needed to remove the tape from the patient’s skin. Cohesion of the PSA was not compromised by the TRAs, as evidenced by the absence of residues on the adherent surface. This novel approach allows rapid introduction into hospital workflows, as heat packs are prevalent in hospital supply systems and are commercially available from multiple sources. This new medical tape (ThermoTape™) requires only a warm compress and its use is not hindered by having the hospital or clinic stock an additional custom device.

We identified two classes of TRA materials which melt within the required temperature range. The first class includes linear long-chain alkanes and their analogous alcohols. C20 (Eicosane) and C22 (Docosane) alkanes exhibit melting temperatures within a desirable range of 38 °C to 43 °C. Furthermore, these alkanes can be combined into an alloy with a melting temperature that is linearly related to the mole ratio of the two adjacent (e.g., C20–C22) alkanes, allowing the temperature to be tuned for different applications. For example, application to children may require a lower temperature than application to adults. Several TRAs were tested in this study, including 1-Tetradecanol (TET), Eicosane (EICO), and a custom, long side-chain acrylic polymer that was synthesized in our lab. This semicrystalline polymeric additive is a copolymer of C14-alkyl acrylate/C18-alkyl acrylate. Different molar ratios were synthesized to achieve a desired melting temperature [18]. ThermoTape™ is formulated with TRA dissolved in an ISO 10993 compliant hybrid acrylic-rubber pressure-sensitive adhesive (LOCTITE DURO-TAK AH 115 Henkel Corporation). Varying amounts of these additives were used in the PSA, and the resulting prototype tape was tested by peel testing and atomic force microscopy (AFM). The PSA manufacturing conditions (wet film drying times and temperatures) were also varied and characterized using peel testing and AFM. The resulting ThermoTape™ prototype may be the first practical medical tape able to significantly reduce the risk of MARSI while simultaneously providing the high adhesion required for secure attachment of medical devices during hospital care.

## 2. Results

Semicrystalline copolymers of C14-alkyl acrylate/C18-alkyl acrylate were synthesized with four different molar ratios, and their melting points were measured with DSC alongside pure C14 and C18 (homopolymers) to demonstrate the tunability of the melting temperature. Figure 1 shows the melting temperatures of the copolymers. Because the 40.45 °C peak of the 73%-C18 and 27%-C14 copolymer was within the temperature range of interest for adhesive removal from skin, this copolymer molar ratio was chosen as the TSP for incorporation into the PSA.

TSP was synthesized as described in Materials and Methods. TET and EICO were purchased from Sigma-Aldrich and used without further purification. The TRAs were mixed with AH-115 PSA at 1 and 5 percent (*wt*/*wt*%) and the resulting coated adhesives were peel tested on LDPE substrates. Figure 2 shows that 1% TSP had the largest reduction in peel strength (76%) from 25 °C to 45 °C, when compared to the other percentages and additives. Next, several percentages around 1% were tested, shown in Figure 2b, and 1% TSP again had the largest decrease in peel strength when heated from 25 °C to 45 °C.

Several drying temperatures and durations were tested with 1% TSP, with the resultant peel strength data shown in Figure 3. From this data, it can be seen that oven drying (solvent evaporation) wet films with 1% TSP at 120 °C for 10 min showed the largest drop in peel strength when heated from 25 °C to 45 °C. This sample was then peel tested at 5 °C increments from 25 °C to 45 °C, compared to pure PSA with no additive and pure C14 and C18 as 1% additives, as shown in Figure 4.

In addition to peel testing, all samples were characterized using AFM. While no TET or EICO samples showed clear surface segregated domains on the PSA exterior (air side), all TSP samples exhibited phase-separated nanometer-sized domains. The formation of these domains was apparently dependent on TSP concentration as well as drying time and temperature. As shown in Figure 5a–c, the domains increased in size with the increase in additive concentration. Furthermore, Table 1 shows that the domains increased in size and overall area when drying time was increased from 10 min to two hours, and temperature produced the same effect when increased from 80 °C to 120 °C. The size of these phase-separated nanometer-sized domains directly relate to the drop in peel strength with heating, where sizes closer to 30 nm showed the largest peel strength drop.

## 3. Discussion

DSC measurements of TSP confirmed that compositions with 27%-C14 and 73%-C18 had a peak melting temperature of 40.45 °C, within our target range for skin adhesive release. These spectra show that the TSP additive can be predictably tuned for each specific use case, such as synthesizing a lower temperature for neonates and an elevated temperature for young adults in outdoor environments.

The peel strength measurements under temperature-controlled conditions demonstrate that the 1% TSP additive provided optimal performance of high adhesion at skin temperature of 35 °C and significant release at 45 °C, as shown in Figure 4. In comparison to pure PSA, the addition of 1% TSP retained 89% of the peel force at 35 °C. From 35 to 45 °C, the pure PSA reduced adhesion by 17.7%, and formulations containing 1% TSP reduced adhesion by 67.5%. This temperature-responsive reduction in peel force was due to the combined synergistic temperature response of the PSA and the additive. Therefore, the combination of the two materials demonstrated a large temperature-triggered drop in peel force while retaining a high initial adhesion. Although these relative differences were promising, the 50-micron PET backing used in these measurements was not representative of a clinically relevant tape for human subject testing, because it is not flexible and its water vapor transmissive properties are too low.

AFM phase images of the PSA containing a long alkyl chain alcohol (TET) or a long alkyl chain alkane (EICO) did not exhibit the phase-separated surface domains which were clearly seen in the TSP samples. Furthermore, mixtures of the commercial AH-115 PSA with the alkane or alcohol additives did not show a strong temperature-dependent reduction of peel strength compared to the 1% TSP additive made with 27% C14 and 73% C18 as a copolymer, as shown in Figure 2. Based on the AFM images and peel strength measurements, it appears that the long chain alkanes or alcohols did not phase separate, and they remained miscible in the PSA following solvent extraction. In the TSP samples, these nanodomains showed a different phase from the surrounding PSA material. We assume that these nanodomains were the TSP additive, as the concentration and size of the nanodomains were dependent on the amount of additive, as shown in Table 1 and Figure 5. As the size of the nanodomains grew with a shorter drying time and higher temperature, it is likely that there was still TSP in the bulk of the PSA film upon removal from the oven. As the film was a given longer time to dry at a lower temperature, the solvent evaporated more slowly, driving more TSP to the surface thereby yielding larger surface nanodomains when drying was complete. With a larger TSP percentage, the domains were larger as there was simply more material, as shown in the AFM images in Figure 5. This demonstrates that the peel strength drop upon application of heat was dependent on the size of the nanodomains. Generally, prototypes exhibiting domains at largest sizes yielded a smaller drop in adhesion upon application of heat, while smaller sizes with sufficient surface area or concentration exhibited a larger decrease in peel strength under our test conditions [19].

The hybrid acrylic–rubber PSA is a highly phase-separated material where the acrylic backbone has significant hydrophilic character (hydroxyl groups (-OH)) to facilitate moisture vapor transmission. The grafted rubber chains and the tackifier exist as a separate hydrophobic phase that is compatible with the TSP. During solvent evaporation, nucleation of the semicrystalline polymer takes place in the bulk PSA. It is highly likely that the semicrystalline polymer has decreased solubility in the solvent-free PSA and migrates along with the evaporating solvent to the exterior (air side) of the adhesive film. A similar mechanism was described when surface modifying macromolecules (SMMs) were blended into polymer solutions [20]. During the process of casting a polymer solution into a film and solvent removal by evaporation, SMMs migrated to the membrane surface. Since the TSP material is not chemically bound into the film it follows the movement of the solvents to the air interface, which adds to its surface enrichment. In addition, preferential surface segregation of the low surface energy TSP is thermodynamically driven to decrease the interfacial energy and minimize the overall free energy of the PSA [21,22]. Thus, surface migration leads to the growth of TSP nanodomains on the PSA exterior surface. The proposed mechanism for the nanodomain formation process is shown in Figure 6. The growth of these nanodomains is affected by TSP concentration, the drying duration, and the temperature.

There are two likely mechanisms for the temperature-responsive release behavior of the TSP additive. In one scenario, upon application of heat, the surface nanodomains melt and coalesce to form “puddles” of non-adhesive material, which disrupts the adhesive–adherend interface. These puddles may also act as stress concentrations that initiate tape detachment upon tape removal. These stress concentrations, or failure points, allow the tape to be removed with less force, acting as nucleation sites for cavity formation during the debonding process. Several publications have described the dynamics of PSA release from a surface, which includes the formation of small cavities at or near the PSA–surface interface as the PSA is pulled from the surface. As pulling continues, these cavities expand and merge into fibrils, defining the final stage of release. Stress concentrations could enable cavity formation at lower forces [23,24,25,26]. Additionally, it is expected that the concentration of TSP exists in a gradient towards the surface. 

Alternately, as the surface and near-surface TSP melts, it increases polymer mobility and decreases the resistance of the PSA to deformation, enabling cavity and fibril formation to occur at a lower force during the deformation and debonding process. In either case, the size of the surface domains is a controlling factor in the temperature-responsive behavior of the adhesive. We expect that smaller nanodomains melt quicker and at a lower temperature than larger ones; the 0.25% TSP sample had smaller domains, and we assume that the concentration was not high enough to sufficiently disrupt adhesion upon melting. 

As seen in Figure 4, most of the peel strength drop occurred after 40 °C, where the TSP melted and disrupted adhesion. The temperature-dependent peel force curve of the pure PSA showed a modest temperature sensitivity. However, there was an abrupt temperature response for the formulation with 1% TSP, close to the melting temperature of the TSP. C18, with a melting point of 50 °C, mostly followed the pure PSA curve until it neared 50 °C. On the other hand, the C14 polymer has a melting point close to 25 °C and behaves as an oily surface residue, and so remained at low adhesion levels starting from 25 °C. Adding 1% TSP slightly lowered the initial adhesion at 25 °C compared to pure PSA, due to the non-adhesive additive on the PSA surface. The 5% TSP additive produced an even larger impact on the initial adhesion. After optimizing a tape system by selecting the base PSA, and the optimal temperature-sensitive additive, concentration, and drying conditions, there remain several limitations to these study findings.

A limitation in the development of ThermoTape™ is the method of heat application with the peel tester. When heating a tape sample to 25 °C from 45 °C, it can take the substrate 10 min to reach the target temperature, subjecting the tape sample to a 10-min heat ramp. Once the target temperature is reached, the tape sample undergoes static heating conditions for one minute before the peel test is initiated. Future clinical use will require more rapid warming conditions, in which a heat pack is applied to the tape for a short amount of time. Additionally, in vitro testing used a heat source maintained by a PID controller, resulting in a consistent target temperature. However, in vivo testing will apply a pre-heated heat pack to tape on skin, with blood flow beneath the tape sample acting as a heat sink that will make heating the tape sample and maintaining the target temperature difficult. In future studies a much thinner and more flexible backing material will be used, which should allow more rapid heat transfer. Given these differences between in vitro and in vivo testing, the brand of heat pack used and the heat pack application time will require definition using pilot clinical studies. Furthermore, the surface migration of the TSP additive during solvent evaporation is a dynamic process. Dynamic measurements will provide more accurate results, which can be used to verify future analytical models of the migration of the TSP additive during the drying process, and of TSP melting during heating. Another limitation lies in the consideration of the drying temperature and duration. We have demonstrated the relationship between nanodomain growth and TSP concentration, drying temperature and duration. However, the drying temperature and duration used in this study are only suitable for low-volume prototyping in static drying ovens. Large-scale manufacturing will use a roll-to-roll (R2R) process at an adhesive coating commercial facility. R2R manufacturing of PSA medical tape utilizes heated forced air solvent evaporation with much shorter drying times. These R2R coaters are multi-zoned, and are cooler in the first zone to remove solvents with low boiling points, continuing in a series of zones that get hotter as they progress, eventually reaching ~120 °C. These ovens have significant airflow to further accelerate solvent removal, differing from the static ovens used in this study. Given the clear dependence on drying time and temperature of the TSP tape, this poses a unique challenge for ThermoTape™ to optimize a pilot manufacturing process that consistently yields the desired nanodomain size without compromising R2R speed.

To provide a more representative prototype for human case study, a more flexible 4.5-micron thick PET backing (PolyK Technologies, State College, PA, USA) was used for testing 1% TSP on skin. Three pilot studies were conducted. In the first study with four volunteers, the coated sheets were cut into 1 × 3 inch samples, with a two-part liner system to allow easy application to skin, as shown in Figure 7. Before applying the tape, both forearms were cleaned using isopropyl alcohol wipes and skin markers were used to mark the general locations of the tape, with two inches of spacing between each pair of lines. Subsequent to pre-tape application preparation, 1 × 3 inch Tegaderm™, Durapore™, and ThermoTape™ were applied in vertical succession on both arms. After each application, the tape was briefly rubbed in a vertical motion to ensure that it fully adhered. After all the tapes were applied, subjects were given an activity log and requested to not participate in strenuous physical activity for the duration of the 24-h study. When subjects returned to have the tape removed, they were given a Wong-Baker 1–10 Pain Rating Scale to fill out after each piece of tape was removed. Prior to tape removal, images were taken to document any redness, allergic reactions, or inflammation that occurred from 24 h of wear. None of the four volunteers exhibited any of these issues with the tapes in the 24-h period. When removing the tape, an edge was lifted and peeled 180 degrees at a consistent rate. If a hair was encountered, the tape was removed following the root of the hair to the tip. On one forearm, all three samples were removed without heat, and on the other, a Dynarex Instant Hot Pack was used on all three samples. The heat pack was activated and kneaded for one minute before application onto the subject’s skin. In preliminary testing, the heat pack was applied to each piece of tape for 10 s before being removed. The data in Figure 7 below illustrates that, without heat, the removal of ThermoTape™ was more painful than Durapore™ or Tegaderm™. Durapore™ and Tegaderm™ exhibited slightly increased pain levels when heated before removal, while ThermoTape™ showed a 65.6% drop in pain levels, which is close to the 67% drop in adhesion when warmed to 45 °C in vitro. A previous study (Krejsa, et al.) showed a correlation between peel force and a subject’s perceived pain level after 24 h of wear, granting significance to a 24-h pilot study [27]. The decrease in pain levels with heat provides initial in vivo validation that ThermoTape™ can reduce pain upon removal when heat is applied. 

The second pilot study used the two volunteers that had previously exhibited the greatest pain range with tape removal. Given that with in vitro testing the tape was exposed to a heat ramp for 10 min, longer heat pack application times were tested to investigate whether pain levels could be further reduced, as ThermoTape™ removal with heat was still not painless. A previous study on young adults reported that heat application for 60 s at 43 °C caused long lasting cutaneous hyperaemia in forearms, which maintained a 5 °C increase in skin temperature for 15 min [28]. Following a similar procedure to the first pilot, five pieces of 1 × 2 inch ThermoTape™ were applied to the forearm. For removal, the heat pack was applied for 0, 10, 20, 30, and 60 s. Once again, no obvious redness, inflammation, or allergic reactions were observed prior to tape removal after 24 h of wear. For the first subject, the pain levels progressed as 5, 3, 4, 2, 2, and for the second subject, as 4, 1, 2, 0, 0. This hints that 30 s of heat application could further reduce pain upon removal, when compared to shorter heat application times. 

The third pilot test involved two volunteers. Strips of 1 × 2 inch ThermoTape™ and Tegaderm™ were applied to the forearm of both volunteers. Images of the forearm were taken at 1, 24, 48, and 72 h after application to investigate possible redness, inflammation, or allergic reactions that may stem from longer-term use of ThermoTape™. In one volunteer there was no redness observable in any tape at any time. In the second volunteer color images were sent to a consulting registered nurse, who was blinded to the type of tapes. Using the erythema and dryness grading scales of Farage (2000) [29], there were no values higher for ThermoTape™ than Tegaderm™ in any of the images. The LOCTITE DURO-TAK AH 115 PSA used in ThermTape™ meets the ISO 10993-5 and -10 standards for skin contact, so the risk is low, with any potential adverse reactions stemming from the 1% TSP additive. While initial testing with 72 h of skin contact indicates that ThermoTape™ is safe for skin use, future work will include biocompatibility testing.

Future clinical trials will use the prototype ThermoTape™ on the 4.5-micron PET backing, since full day wearability has been proved easily achievable and estimated moisture vapor transmission is acceptable for experimental human use. Future ThermoTape™ development and clinical testing in hospital settings will use higher moisture vapor transmission backing materials such as polyurethane and will be tested after sterilization. 

## 4. Materials and Methods

### 4.1. Polymer Synthesis

All polymers used in this work were prepared using the same copolymerization procedure described in detail in a prior publication [18]. The tetradecyl acrylate (C14) monomer (KPX Chemical Co., Seoul, South Korea) was purified over neutral alumina, while octadecyl acrylate (C18) monomer (KPX Chemical Co., Seoul, South Korea), azobisisobutyronitrile (AIBN) initiator (Sigma Aldrich, St. Louis, MO, USA), and toluene were used as received. Initiator and solvent concentrations were held constant at 0.2 and 65 wt%, respectively, while the copolymer composition was controlled by adjusting the molar ratio of C14/C18 monomers. The following serves as an example synthesis procedure:

AIBN (0.02 g, 0.12 mmol) and C18 (7.82 g, 24.18 mmol) were added to a 50 mL round bottom flask. C14 (2.18 g, 8.12 mmol) and toluene (3.0 g) were added to a separate scintillation vial, homogenized with a vortex mixer, and added to the reaction flask. The vial was rinsed with additional toluene (3.5 g) to ensure complete addition of the C14 monomer. The mixture was allowed to dissolve, and sparged with N_2_ gas for 15 min. The polymerization reaction was performed in a small, heated (85 °C) round-bottom flask immersed in a sand bath for 18 h. The mixture was precipitated into ethanol and the copolymer was collected by vacuum filtration. The copolymer was redissolved in a minimal amount of toluene, precipitated once more in fresh ethanol, and dried in a vacuum oven at 60 °C for 24 h to remove trace solvent contamination. 

### 4.2. Differential Scanning Calorimetry 

Polymer melting points were determined using a TA Discovery DSC 2500 differential scanning calorimeter (DSC). All samples were initially heated to well above their melting temperatures (T_m_) at 20 °C/min, cooled below their T_m_ at 1 °C/min, and reheated at 10 °C/min. All thermal characteristics reported in this work were taken during the second heating.

### 4.3. Nuclear Magnetic Resonance (NMR)

All compositions were verified using a 499 MHz Bruker Avance III spectrometer, the terminal methyl group for calibration, and the secondary side-chain hydrogens for quantitative comparison.

### 4.4. Analytical Tools

#### 4.4.1. Atomic Force Microscopy (AFM)

Surface morphology of the PSA dried coatings was analyzed by AC-mode phase contrast AFM. Images were taken with an Oxford Instruments Asylum Research Jupiter XR AFM (Asylum Research, Santa Barbara, CA, USA) using BudgetSensors Tap300Al-G tips. The cantilever was tuned to a free amplitude of 300 mV and operated at a set point optimized to obtain the highest possible average phase, ensuring the images were obtained in attractive mode at a phase well above 90°.

#### 4.4.2. Peel Testing

Peel testing was conducted using a test apparatus constructed based on Test Method F of ASTM D 3330/D 3330 M [14,19]. Adhesive tape was applied to a temperature-controlled low-density polyethylene (LDPE) substrate after it reached a target temperature, and was left for one minute before initiating the test. The tape was secured to the substrate using a ChemInstruments RD-1000 rolldown machine with a 4.5-pound roller (ChemInstruments, Fairfield, OH, USA). A standard peel rate of 100 mm/minute was used at a peel angle of 180°. Tape samples were tested at numerous substrate temperatures. The peel force data were analyzed with MATLAB (Version R2022a, The Math Works, Inc., Natick, MA, USA).

### 4.5. Prototype Tape Fabrication

Three different TRAs were tested separately at various *wt/wt*% mixtures of TRAs and PSA, with TRAs of the 73%-C18 and 27%-C14 copolymer temperature-sensitive polymer (TSP), 1-Tetradecanol (TET) (Sigma Aldrich, St. Louis, MO, USA) and Eicosane (EICO) (Sigma Aldrich, St. Louis, MO, USA). The solid additives were first added and dissolved in the solvent-based PSA by stirring for 24 h at room temperature. This solvent-borne mixture was subsequently deposited onto a pre-cleaned 50-micron thick polyethylene terephthalate (PET) clear backing using a slot die sheet coater (FOM Technologies, FOM alphaSC, Copenhagen, Denmark). This process was performed in a controlled environment with humidity and temperature set at 40% and 70 °C, respectively. The resulting coated sheet was then placed into a drying oven for a set time and temperature to provide varied drying conditions.

## 5. Conclusions

ThermoTape™ is the first temperature-sensitive tape that functions in the human skin temperature range. In vitro testing has demonstrated retention of high adhesion from 1% TSP at normal skin temperature (35 °C), and 67.5% reduction in peel force adhesion when raised to 45 °C. Initial in vivo testing has shown a 65.6% reduction in pain when heat is applied prior to removal. We believe that successful translation will eliminate the current decision dilemma that nurses must make every day; either to maintain the highest quality care by using high-adhesion medical tape which risks MARSI upon removal, or to avoid MARSI by using lower-adhesion tape more likely to result in medical error and device dislodgement. The importance of a medical tape with optimal properties is key to patient satisfaction, and is dependent on all aspects of treatment, including patient discharge when medical tape is removed.

## Figures and Tables

**Figure 1 ijms-23-07164-f001:**
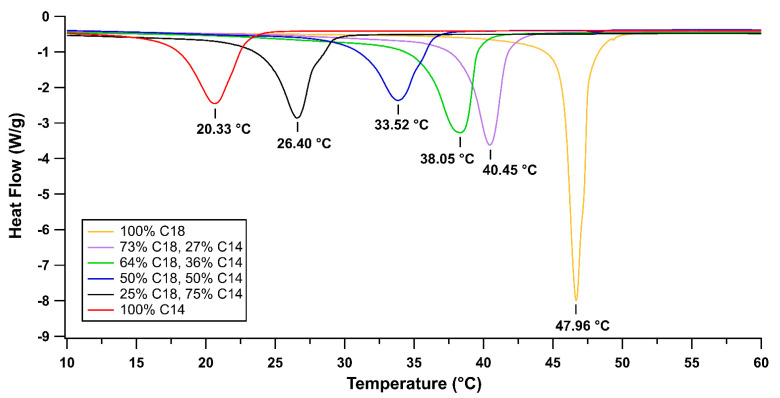
DSC curves for C14 and C18 homopolymers and four C14/C18 copolymers, demonstrating the tunability of the C14/C18 copolymer with varying C14 and C18 molar ratios.

**Figure 2 ijms-23-07164-f002:**
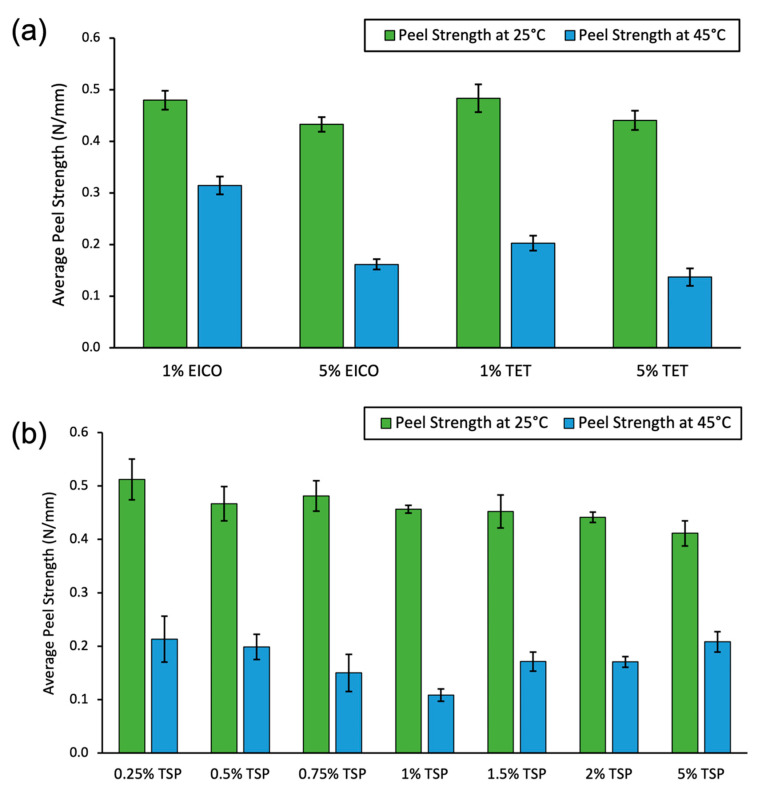
Peel strength data for (**a**) 1% EICO, 5% EICO, 1% TET, and 5% TET tested at 25 °C and 45 °C, and (**b**) C14/C18 copolymer TSP percentages at 25 °C and 45 °C.

**Figure 3 ijms-23-07164-f003:**
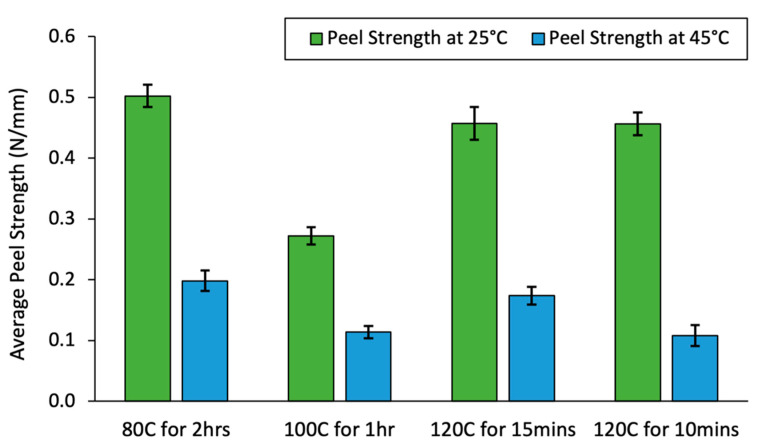
Peel strength data for 1% TSP for varying drying times and temperatures, tested at 25 °C and 45 °C.

**Figure 4 ijms-23-07164-f004:**
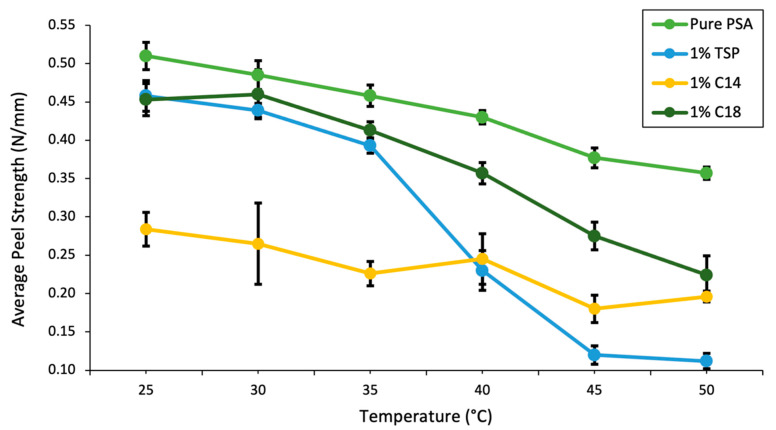
Peel strength curves for pure PSA with no additive, 1% TSP, 1% C14 homopolymer, and 1% C18 homopolymer. Materials were tested at 25–50 °C in 5° increments.

**Figure 5 ijms-23-07164-f005:**
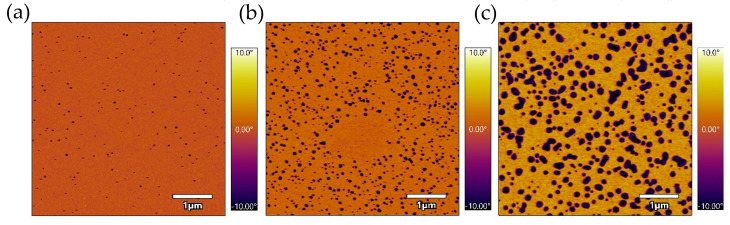
AFM phase images of 0.25% TSP (**a**), 1% TSP (**b**), and 2% TSP (**c**). Areas of low phase (dark) correspond to TSP nanodomains.

**Figure 6 ijms-23-07164-f006:**
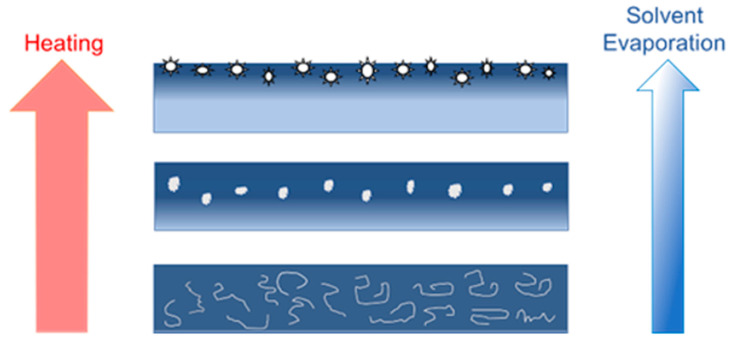
Hypothesis for TSP nanodomain formation on the ThermoTape™ surface. Initially, TSP is dissolved in the PSA solvents. As the PSA and additive are heated, the solvent evaporates, causing the TSP to begin nucleation and growth. The movement of solvent to the air interface carries the growing TSP particles towards the surface and preferentially deposits them on the PSA surface. Upon application of heat, these surface TSP nanodomains melt to form non-adhesive puddles on the skin-PSA interface, providing numerous stress concentrations to initiate tape delamination at a lower peel strength. Alternatively, the melted TSP facilitates PSA cavitation and filament formation during tape removal.

**Figure 7 ijms-23-07164-f007:**
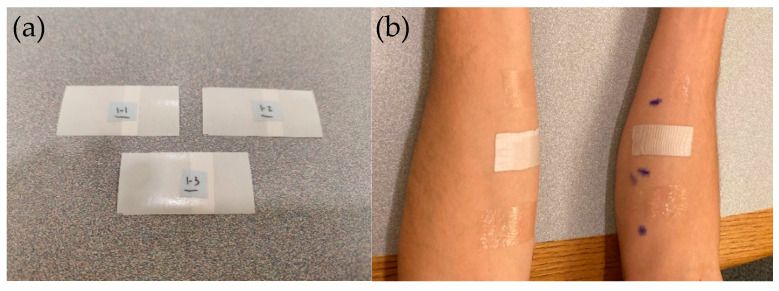
Prepared 1 × 2 inch 1%TSP ThermoTape™ prototypes for pilot clinical testing (**a**). 6 samples before removal: Tegaderm™ at the top, Durapore™ in the middle, and ThermoTape™ at the bottom. No redness is visible, indicating no obvious allergic reactions and inflammation during 24 h of wear (**b**). Pain level data for Durapore™, Tegaderm™, and ThermoTape™ with and without heat (**c**).

**Table 1 ijms-23-07164-t001:** TSP nanodomains’ increase in size with increased TSP percentage, drying time and drying temperature.

Sample	Average CE Diameter (nm)	Average Area (nm^2^)
0.25% TSP 120 °C 10 min	8.137 ± 13.45	193 ± 350
1% TSP 120 °C 10 min	30. 155 ± 28.389	1350 ± 1820
1% TSP 100 °C 1 h	78.237 ± 22.882	5220 ± 2930
1% TSP 80 °C 2 h	77.015 ± 26.918	5230 ± 3410
2.0% TSP 120 °C 10 min	130.989 ± 60.985	16,400 ± 14,700

## Data Availability

The data presented in this study are available on request from the corresponding author. All data are not publicly available due to human testing restrictions.

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
