# Peer review of "Prototype Development of a Temperature-Sensitive High-Adhesion Medical Tape to Reduce Medical-Adhesive-Related Skin Injury and Improve Quality of Care"

_ijms, 2022, doi:10.3390/ijms23137164_

Round 1

Reviewer 1 Report

This manuscript talks about skin lesions related to medical adhesives, in the literature it is a recent topic, worthy of study. There are studies related to injuries of mechanical origin. In these cases the authors of these works suggest to invest in preventive measures and good clinical nursing practices. This article presents a new approach in the prototype development of a temperature sensitive. I agree with the publication of this manuscript, but after completed with a study of possible effects that may occur in the use of this medical tape, effects such as irritation, allergy and inflammation.

Author Response

“Reviewer 1 stated that the introduction, research design, and conclusion could be improved.  Specifically, a study of possible effects that may occur in the use of this medical tape, effects such as irritation, allergy and inflammation.

Thank you for these comments and suggestion for expanding our scope of work.  We have added results of a new (third) pilot study for longer duration of wear, which was 72 hours rather than just 24 hours. Clinically relevant results were scored by a registered and experienced nurse and co-author (A.T.) according to procedures cited in the added reference [29] Farage (2000). In the Discussion, additional comments were included for any obvious redness and irritation (which there were none) for the short- term studies of 24-hours wear. Clarification was made that the 24-hour studies were designed to better match the pain scores, as referenced by the new [27] reference.

In addition, the introduction was edited for improved readability and understanding. As stated above with the addition of the third pilot study (72-hours wear) results and clarification of the first two pilot studies (24-hours wear) were made with two additional references. These additions improved the research design and conclusion as requested by the reviewer.”

Reviewer 2 Report

The development of a high adhesion medical tape designed for low skin trauma upon release is reported in this manuscript. A C14/C18 copolymer was developed and combined with a selected pressure sensitive adhesive (PSA) material. The adhesive film was characterized with AFM, where distinct nanodomains were identified on the exterior surface of the PSA.

The scientific content of the ms is very interesting and I thus favor its acceptance in IJMS. I am sure that the paper will attract the intense interest of scientists working in the area of the development of medical devises. Also, I do believe that the article will receive a respectable number of citations in the future. Salient features of this work – which support my proposal for acceptance – are the development of  ThermoTape™ a  temperature-sensitive tape functioning in the human skin temperature range. In vitro testing has demonstrated retention of high adhesion from 1% TSP at normal skin temperature (35 °C), and 67.5% reduction in peel force adhesion when raised to 45 °C, while Initial in vivo tests have shown a 65.6% reduction in pain when heat is applied prior to removal. The quality of figure is high and the references list covers the topic under study satisfactorily.

Author Response

Reviewer 2 comments were all positive which indicated that there would be great interest and quality is high.  Only the references were listed as satisfactory.  In response to this reviewer and subsequent reviewers, two additional references are added to explain and support the testing procedures used in this manuscript. Thank you for these supportive comments. 

Reviewer 3 Report

The paper is interesting. It treats a very important topic of medical adhesives. I suggest that the introduction be revised and written better. To take into account the abbreviations when and where they are written for the first time. I also suggest that the results and discussion be compiled.

Author Response

Reviewer 3 stated that the introduction, cited references, and results could be improved. Specifically, the introduction should be revised and written better, and the abbreviations should be introduced more appropriately.  Also, the reviewer suggested that the results and discussion be compiled or combined, the editor’s email clarification.

The introduction was heavily edited and improved for readability. The abbreviations were explained more clearly throughout the revised manuscript. Two additional references were added to explain and support the testing procedures used in this manuscript. Although there are pilot study results in the discussion section, combining the results and discussion sections would make for a very long section and maybe too long for readers. We can agree with this recommended combination of results and discussion only if the journal editors agree, since the other two reviewers did not see this as a needed correction.

Round 2

Reviewer 3 Report

I agree with the corrections made. Now the paper is at a great level.